# Exploring joint decision-making and family dynamics to identify barriers and enablers for early adolescent medical circumcision (EAMC) uptake in Zambia for HIV prevention: An innovative methodology

Rasi Surana[1], Ram Prasad[1], Namiya Jain[1], Mothi Prasad[1], Alok Gangaramany [1], Aishwarya Shashi Kumar[1], Tim Sweeney[2], Jeff Mulhausen[2], Steve Kretschmer[3], Alick Samona[3], Alice Nanga[3], Tina Chisenga[4]

1 Final Mile Consulting, New York, New York, United States of America, 2 Upstream Thinking, Austin, Texas, United States of America, 3 DesireLine, Istanbul, Turkey, 4 Ministry of Health, Lusaka, Zambia

## Abstract

Voluntary medical male circumcision (VMMC) to protect against sexual transmission of HIV is a key part of HIV prevention interventions in 15 priority countries in Southern and Eastern Africa. Ensuring that VMMC programs reach adolescent males is important in countries with large young populations. We designed a methodology to explore the joint decision-making dynamics among caregivers and adolescents aged 10–19, and the drivers and barriers for circumcision, in order to identify levers which can drive uptake of VMMC. Our approach was grounded in behavioral science to address some of the limitations of survey-based research (e.g., the "say-do gap," social desirability bias, respondent fatigue). Our methods included 1) interviews with adolescent boys and their caregivers to understand how adolescents interact with their families, other key stakeholders, and the healthcare system; 2) journey mapping to understand how boys and caregivers move through the stages of progress toward the decision for VMMC, and the influence of context, family, and community members; and 3) Ethnolab, a decision-making game that tests behavioral hypotheses in hypothetical situations mimicking the real-life context of decision-making about VMMC, enabling an understanding of boys' and caregiver's motivators, barriers, and mental models via observation as well as questioning. Factors influencing the decision for VMMC included anticipated pain of the surgical procedure, mistrust about safety, the boy's uncertainty about his caregiver's consent, and caregiver's uncertainty about the adolescent's assent, and caregiver's concern about their adolescent boy's maturity level and ability to deal with VMMC, among others. Conversely, in-group seeking, the belief that being circumcised is appreciated by women, and improved hygiene were among the positive factors motivating decisions for VMMC. Demand generation should involve the

**Data availability statement:** All relevant data are within the manuscript and its Supporting Information files.

**Funding:** Author: AG Funder: Bill & Melinda Gates Foundation Investment ID INV-003302 https://www.gatesfoundation.org/ The funder played no role in the study design, data collection and analysis, or preparation of the manuscript.

**Competing interests:** The authors have declared that no competing interests exist.

whole family unit, encouraging discussion and trust within and among households, and recognizing and addressing the ways decision dynamics change as the boy ages through adolescence.

## Introduction

Since the efficacy of voluntary medical male circumcision (VMMC) in conferring protection against sexual transmission of HIV from women to men was demonstrated more than 15 years ago [1–3], VMMC has been a key part of HIV prevention interventions in 15 priority countries in Southern and Eastern Africa as identified by UNAIDS & WHO [4]. By 2020, an estimated 615,000 new HIV infections had already been averted through the 29.5 million VMMCs performed [4].

In 2020, 79% of VMMCs conducted in 10 of the 15 high-priority countries were among males aged 10–24 years, with a large proportion of them being adolescents aged 15–19 [4]. VMMC is a one-time intervention, unlike Pre-exposure prophylaxis (PrEP) or condoms, that significantly reduces the risk of HIV transmission and other sexually transmitted infections (STIs) [1–3]. It is particularly relevant for adolescents as the procedure offers the greatest protection if it is done before sexual debut, as it reduces the lifetime risk of acquiring HIV. Therefore, targeting adolescent boys who have not yet become sexually active maximizes the health benefits of VMMC. For countries such as Zambia which are experiencing a youth bulge (in 2022, an estimated 24% of the population was aged 10–19) [5], ensuring that VMMC programs reach adolescent males is therefore critical to successful HIV prevention.

In 2020 the World Health Organization (WHO) revised the age criteria for VMMC programs for adolescents (also referred to as Early Adolescent Male Circumcision) to 15–19 years [6] because of concerns about 10–14 year old adolescents' ability to give informed assent to a medical procedure that is non-urgent and irreversible and due to a higher rate of serious adverse events following Male Circumcision (MC) among this age group. In line with this updated guideline, Zambia's VMMC program now focuses on reaching uncircumcised males aged 15 and above [7].

While EAMC is a collective decision involving the adolescent boy and his caregivers, most current interventions to drive demand for EAMC do not address key questions about the decision dynamics between adolescents, especially those under the age of 18, and their caregivers. EAMC for boys under 18 requires both the adolescent's assent and the caregiver's consent [8], as MC could the adolescent's choice acquiesced to by his caregiver, a collective decision by the caregiver and adolescent, or a choice that caregivers make for the adolescent boy [9, 10]. Differences in understandings, motivations, and beliefs about EAMC between the boy and his caregiver, and varying individual maturity informing the adolescent's ability to assent to EAMC, all add an additional layer of complexity to the decision-making dynamic.

The present paper describes a methodology specially designed to address this complexity underlying decision-making related to EAMC adolescents and their caregivers in Zambia. By focusing on the joint decision-making dynamics within families and identifying the drivers and barriers to circumcision for adolescents, our

methodology provided unique insights into how interventions could be more effectively tailored to increase EAMC uptake. Our study included younger adolescents (10–14 years in addition to 15–19 years) as the study design and initial fieldwork was completed prior to Zambia's alignment with the revised WHO guidelines promoting VMMC for those aged 15 and older [6]. This paper details the theoretical foundations and qualitative research methods we employed, highlighting how these approaches successfully addressed limitations in traditional research methodologies. Ultimately we aim to provide actionable insights and a behavioral decision-making framework for improving VMMC uptake (a more detailed account of which will be published separately).

## Background

Cultural context, social norms, and interpersonal dynamics between adolescents and their caregivers can create barriers to decision-making for EAMC. Among non-circumcising tribes for instance, where ethnic identity is associated with a negative perception of MC, the absence of circumcision traditions often leads caregivers to deny permission to their adolescents, or to not consider the option of circumcision altogether [11–13]. In these provinces, scaled coverage of VMMC is not possible unless positive messages about it are disseminated by traditional leaders. Caregivers and young people report several barriers to open dialogue, including lack of knowledge and skills, as well as cultural norms and taboos [14, 15]. There are also differences in caregiver engagement: while both female and male caregivers contribute to the decision, they state that VMMC is primarily a matter for male heads of households to address [8, 9].

In addition to social and cultural challenges, EAMC uptake has some structural barriers as well, logistical challenges such as distance to the medical facility, lost wages during the recuperation period (for those who work), and inconvenient timing, as either because services are often unavailable outside school or work hours, or coincide with school exam periods or sports tournaments [16–20]. In addition, poor-quality services, an environment unwelcoming to adolescents, and concerns about violations of privacy can serve as further deterrents to EAMC uptake [10].

Conversely, several factors have also been found to facilitate uptake of EAMC. Beliefs that MC reduces the risk of HIV infection, improves hygiene, and enhances sexual desirability and satisfaction are prevalent [16–18]. Previous research suggests that motivations for choosing EAMC vary among adolescents of different ages - for instance, in comparison with adolescents aged 15–19 years, those aged 10–14 are less likely to seek EAMC for protection from HIV or other sexually transmitted infections, or for hygiene reasons, instead, they are more motivated by advice from others [21]. Anticipation of shame and stigma also changes with age, with younger adolescents less concerned than older ones about the potential stigma of being uncircumcised [22]. Despite these varying motivations, adolescents share some cognitive barriers to EAMC including fear of pain, and the fear of stigma associated with potential HIV-positive results, as an HIV test is typically required prior to the procedure [18,21,22].

The capacity of adolescents to give informed assent is also crucial. Research shows that while a slightly higher proportion of adolescents aged 13–17 than of adults passed a comprehension assessment of key concepts related to VMMC, but adolescents scored significantly lower than adults on two questions related to the risks of surgery and whether all circumcised men are HIV negative [8,11]. Age-related differences in risk tolerance and decision-making processes add further complexity. Adolescents aged 14–17 have the same cognitive and reasoning capacity as 18–19-year-olds. However, the younger adolescents have a higher tolerance for risk, are easily influenced by peers/caregivers, and attend primarily to short-term consequences of their actions [23]. The desire for caregiver involvement also differs: a larger proportion of 15–19-year-olds than 10–14-year-olds report being a little or very uncomfortable with caregivers attending their pre-procedure counseling session [8, 9]. A further concern is embarrassment about caregiver involvement in caring for the wound after the procedure, since adolescents – particularly older ones – feel shame about exposing their genitals to their caregivers (who are also embarrassed by this) [9].

There is also evidence for the influence of peer dynamics on health behaviors such as circumcision uptake [24]. Peers play an important role in decision-making for adolescents aged 10–17; for those who believe that their caregivers exert

restrictive control over them, peers may become their primary source of guidance [25]. Younger adolescents also become more concerned about peer acceptance and popularity and begin to turn to their friends more often as sources of advice and comfort [26].

Despite these differences across ages, existing demand interventions also do not specifically address the needs of older adolescents aged 18–19. They are only just maturing out of younger adolescence, and there are reasons to hypothesize that there may not be a complete absence of caregiver involvement after the adolescent turns 18, especially since in Zambia, 62% of 15–19-year-olds (male and female) live in a household with their parents or grandparents, and only 2% of 15–19-year-old male adolescents are heads of their own households (see the analysis of data from the Zambia Demographic and Health Survey 2018 in S1 File).

## Structural limitations of survey-based research

In order to study the aforementioned individual, social, and cultural barriers and enablers to VMMC, most research into the factors hindering or enabling uptake of VMMC uses surveys, one-on-one interviews or focus group discussions. These tools depend on the respondents' ability to reflect and provide a considered and accurate response. However, while people are often good at rationalizing their decisions in hindsight, in practice much of their decision-making process is non-conscious. Decisions are driven by emotions, by mental shortcuts (heuristics) and other non-conscious drivers [27]. These contribute to several potential weaknesses in responses given in surveys, interviews, and discussions. First, the "say-do" gap highlights that in sensitive decision-making contexts, what people say and what they actually do can be poles apart. For example, in a study in Zambia that interviewed 1,000 mothers of newborn boys across two public clinics, 97% of the sample said they probably or definitely would circumcise their newborn son, but only 11% of them brought their son back to the clinic to be circumcised [28]. Second, social desirability bias can lead respondents to answer questions in a manner they think will be viewed favorably by others [29, 30]. Surveys and interviews may create an environment in which respondents feel fearful of being judged and respond accordingly, over-reporting "good/desirable" behavior or under-reporting "bad/undesirable" behavior. For example, Malawian teenagers aged 16–18 were less likely to report ever having had a girlfriend in audio computer-assisted self-interviews than in face-to-face interviews, but more likely to report having had sex with a relative or teacher [31].

Third, respondent fatigue occurs when surveys are too long to maintain the respondent's interest and motivation [32, 33]. As they become tired of the task, their attention declines, and the quality of the data they provide deteriorates. Finally, research methods like focus group discussions are conducted with a sample while they are out of their actual decision-making context, which can affect their responses. A different manifestation of this problem occurs when the context of the people recruited for research is not the same as the wider research context in terms of social norms, mental models, and other contextual factors. For example, a study may recruit and survey university students to draw conclusions about the attitudes and behaviors of people of that age, ignoring the fact that the respondents' age peers outside of university may have quite different social norms and mental models.

## Understanding human decision-making

Individuals' decisions that lead to "say-do" gap or social desirability bias can be deconstructed and better understood through the application of theories from behavioral science. People do not make decisions rationally [34]. They must search for options, evaluate them, and then select among them. This process is not efficient, since decisions are generally made with a non-exhaustive set of options and limited resources of time and mental capacity. This is referred to as bounded rationality [34]. Appraisal theories offer a solution to this process by decoding decision-making and understanding the underlying emotions, which are driven by stimuli (a trigger or a cue in the environment) and people's perceptions of their context. The appraisal of the stimuli within the context generates emotions and leads to action tendencies.

Modern appraisal theories provide a standard set of dimensions that underlie the appraisal process – the relevance, implications, and normative significance of the stimulus, and the individual's capacity to cope with the consequences of the decision [35–37]. Additionally, people also appraise whether stimuli lead to an enhancement of their status (status-seeking), the enabling assistance of their social group (reciprocity striving), and maintenance of strong social bonds (group identity-seeking), as these enable survival [38].

## Methodology

The research was conducted by a consortium supporting Zambia's national VMMC program. There were three components to this fieldwork: formative research, journey mapping, and Ethnolab. The formative research consisted of semi-structured in-depth interviews with adolescent boys aged 10–17 (hereafter referred to as younger adolescents) that, along with previous research undertaken by the consortium [39], informed the design of the journey mapping and Ethnolab. Sampling for these phases was done primarily in 2 districts - the Western District of Sioma and Lusaka district, due to their rural, non-circumcising tradition (Sioma) and urban and cosmopolitan cultural characteristics (Lusaka) respectively. Similarly, the selection of schools and health centers was done based on their geographical location and characteristics, as has been outlined in Table 1 below.

Formative Research and journey mapping were not conducted with adolescents aged 18–19 (referred to hereafter as older adolescents) because the insights gained from a previous study of adults aged 18 and over [39], together with the formative research and journey mapping insights from younger adolescents, was judged to provide sufficient basis to design the Ethnolab research tool for this group. Formative research was completed in June and July 2020 and fieldwork for Ethnolab and journey mapping with younger adolescents was conducted in January and February 2021. Fieldwork with older adolescents took place in December 2021 and consisted only of Ethnolab. Across the 3 phases of research, data was collected from adolescents and caregivers (including parent and non-parent caregivers and single-parent households) to understand the joint-decision making dynamics at play for EAMC decisions.

Ethical approval was received from the ERES Converge review board (Approval number: 2020-Jan-019) in Zambia before each stage of the research, and from Zambia's National Health Research Authority, for both adolescents aged 10–17 and adolescents aged 18–19. Prior to their participation in any stage of the research, an informed consent script was read to each participant in their preferred language. They were given the opportunity to ask questions about their participation. Participants acknowledged their consent by initialing, signing, and writing the date on the consent form (verbal consent was received for some interviews that took place over the phone because of restrictions imposed by the

Table 1. Characteristics of study sites.

| Study Site Type | Specifics | Characteristics |
| --- | --- | --- |
| Primary Public School | New Kanyama Primary School | Urban school based in Lusaka city with mainly cosmopolitan mix of students from all tribes of Zambia. |
| Secondary Public School 1 (attend session) | Chitimukulu Secondary School | Rural school with primarily traditionally connected students from non-circumcising tribe, Lozi. |
| Secondary Public School 2 (interview) | St Monica's Secondary School | Urban school based in Lusaka city which mainly has cosmopolitan mix of students from all tribes of Zambia |
| Primary Community School | Chengelo Community Primary School | Rural school with primarily traditionally connected students from non-circumcising tribe, Lozi. |
| Public Health Care Facility 1 | Kanyama First Level Hospital | Urban, located in Lusaka district. Clients attending the facility are mainly cosmopolitan in nature |
| Public Health Care Facility 2 | Makeni Ecumenical | Urban, located in Lusaka district. Clients attending the facility are mainly cosmopolitan in nature |

COVID-19 pandemic). For illiterate respondents, an agreement to participate in the study illustrated with an inked thumb-print was treated as written consent. An overview of these research phases is presented in Table 2 below.

## Formative research

A sample of 16 adolescent boys aged 10–17 and 22 caregivers were recruited from Lusaka and Chongwe districts for interviews. They were chosen based on the adolescent's circumcision status, age, and school enrollment status (Table 3). In studies utilizing qualitative methods, it is recommended that sample sizes be kept small to facilitate in-depth analysis of participant responses [40]. The sample size for this phase therefore, whilst chosen to be large enough to collect extensive and nuanced information on factors influencing EAMC, was limited to under 40 to allow detailed exploration. Additionally, the concept of 'informational redundancy' or 'data saturation', in which 'gathering fresh data no longer sparks new theoretical insights' [41], proved instrumental in assessing the adequacy of sample size. It was important to sample both in-school and out-of-school adolescents, because while EAMC outreach programs for this age group focus on schools, transition rates from primary to secondary school are as low as 67.5% [42]. Purposive sampling for staff from schools and health facilities was based on their role in promoting VMMC and interacting with adolescents undergoing VMMC.

While restrictions related to the COVID-19 pandemic were in place, recruitment in Lusaka district was done by phone, using a database of contacts available to the research partners. Phone recruitment proved challenging, and after restrictions on movement were relaxed, recruitment was conducted through door-to-door visits and snowballing. In Chongwe, door-to-door recruitment was done in communities where one of the consortium partners had existing relationships. Participants were offered an amount of 100 Kwacha (~$5.00 USD) per household, and 50 Kwacha (~$2.50 USD) for school and facility staff, in lieu of any costs borne by them to participate in the study such as travel expenses, wage loss etc.

**Table 2. Overview of research phases.**

| Research Phases | Time orientation | Sample | Objectives |
|---|---|---|---|
| Formative Research | June and July 2020 | 16 households - 16 adolescents (10–17) and 22 caregivers; 7 health system stakeholders | Exploring the socio-cultural context within which EAMC decisions are jointly arrived at within the household through in-depth interviews with the identified sample. |
| Journey Mapping | January and February 2021 | 36 households - 36 adolescents (10–17) and 72 caregivers | Based on learnings from the formative research, the journey map leveraged participatory research methods to understand the barriers and drivers that adolescents and caregivers face in the EAMC journey. |
| Ethnolab | January and February 2021 (10–17); December 2021 (18–19) | 150 Adolescents (10–19) and 180 caregivers | The Ethnolab built on the learnings of the first two phases to identify actionable levers that help caregivers and adolescents in overcoming barriers and move them forward in their EAMC journey. |

**Table 3. Selection of formative research interview participants.**

| Circumcision Status | Circumcised | | | | Not circumcised | | | | TOTAL |
|---|---|---|---|---|---|---|---|---|---|
| School Enrolment Status | In School | | Out of School | | In School | | Out of School | | |
| Age Group | 10-13 | 14-17 | 10-13 | 14-17 | 10-13 | 14-17 | 10-13 | 14-17 | |
| Number of Adolescents | 1 | 3 | 3 | 1 | 3 | 1 | 2 | 2 | 16 |
| Number of Caregivers | 1 | 3 | 5 | 1 | 5 | 1 | 3 | 3 | 22 |

2 school staff, 2 counselors, peer champions, VMMC coordination, Health Department health promotions officer

Formative research interviews were in-depth interviews, lasting about one hour, conducted by a local field team trained remotely by the consortium. Some interviews were conducted over the phone due to COVID-19 restrictions, with the remainder conducted in person once restrictions were lifted. Interviews were audio recorded.

In the interviews, moderators explored the cultural and social context within which adolescents interact with the health-care system, and more specifically, make decisions with respect to circumcision. Interviews with caregivers explored their relationship with their adolescent boys generally and in the context of circumcision, their interactions with the health system, and their beliefs and traditions. Interviews with school staff and facility providers explored their role, interactions with adolescents and caregivers, and their perspective on the EAMC context.

The interview tool was a discussion guide created for each category of respondent, organized thematically, with the objective and key questions of each theme defined, along with a detailed question bank. Some themes explored were, for example, anticipated loss of wages (while going through the procedure, while caring for the teen), anticipated conflict (when initiating conversation around EAMC). The themes used were identified from our previous research [39], which were augmented, adapted and refined with findings from the formative research with adolescents and caregivers for the EAMC context. The moderators explored the key questions in each theme using the question bank as needed, based on how the conversation was progressing.

Transcripts of the interviews were analyzed using deductive analysis, drawing from Ellsworth & Scherer's emotional appraisal theory [43] and thematic learnings from our previous research [39]. The authors (Namiya Jain, Rasi Surana and Alok Gangaramany) coded transcripts to systematically identify key themes and appraisal dimensions emerging from the data. Data from each household were organized and analyzed together, allowing us to examine interviews across all members of a single household in context of one another. This approach enabled more effective triangulation of insights. After coding, the data were synthesized by respondent category and further grouped by age band to capture nuanced patterns across different demographic groups. These learnings were leveraged to construct hypotheses around the barriers faced and ways to overcome these. The formative research adapted hypotheses from the previous VMMC research for the joint decision-making context of EAMC to arrive at the stages of a household's journey to EAMC (Table 4). This was used as a basis for further stages of research in the study.

## Journey mapping

Journey mapping is a research methodology that analyzes the actual and/or ideal experience of people who have fully or partially achieved a behavioral outcome [44]. It is participatory in nature. It is designed to counter the "say-do" gap through the act of making: the respondent makes their own journey artifact over the course of a journey mapping interview [39].

**Table 4. EAMC journey stages.**

| Stage | Definition |
|---|---|
| Not Intending | Caregivers and the adolescent are aware of EAMC but they do not perceive it to be relevant, and/or anticipate barriers against it. |
| Actively Aligning | The intending member may have barriers around EAMC to be overcome but seeks conversation with the family. However, other member(s) do not perceive its relevance and/or anticipate barriers and are inhibiting action. This stage comprises 2 family types, one where the adolescent is the intending member (A+C-), and another where the caregiver is the intending member (A-C+). |
| Anticipating | EAMC is relevant to both the key decision-maker and the adolescent (A+C+). However, there is an intent-action gap due to perceived barriers by one or both of them. |
| Not Advocating | The adolescent boy is circumcised. However, one or more family members do not encourage others to go for the procedure. |
| Advocating | The adolescent boy is circumcised, and one or more members of the family actively encourage others to go for the procedure. |

Empowering people to make journey artifacts that represent their behavior not only creates a sense of ownership over their story and the process, but also results in a more accurate articulation of behavior and the rationale driving it.

Journey mapping is particularly appropriate in the context of EAMC, since decisions about circumcision are complex and take place over long periods through interactions among multiple people, which include second-guessing, the introduction of new information, and social conflict or conflict avoidance. We therefore used the journey mapping methodology to capture the historical journey of households (son, mother figure and father figure) on their way to EAMC and beyond, to advocating for the procedure.

The journey structure aims to identify behavioral drivers that advance people toward the outcome, and barriers that move people away from it. We used four key milestones as journey stage anchors for specific behavioral probes to understand how they had advanced through prior stages, why they had not progressed through their current stage (if applicable), and what it would take for them to do so:

1. **Desire:** During the Not Intending stage (see Table 3), how household members do or do not move from an awareness about EAMC to at least one member desiring EAMC.

2. **Intent:** During the Actively Aligning stage, how households do or do not move from one member wanting EAMC to a household commitment to get EAMC done.

3. **EAMC:** During the Anticipating stage, how households do or do not progress from a commitment to get EAMC done to actually going through with the procedure.

4. **Advocacy:** Once the household is relieved that EAMC has been done, how household members do or do not evolve to advocating for the procedure with other community members.

A sample of 36 households was recruited (12 each in Lusaka, Sioma, and Western districts), each consisting of 3 respondents: an adolescent male between the ages of 10 and 17, his father and his mother (or father figure and mother figure), for a total of 36 households and 108 respondents (Table 5). This sample size was determined in accordance with previous studies in healthcare research that made use of journey mapping, which have been found to have recruited approximately 71–100 participants [45] as was done in the present study. The samples were evenly divided between 4 of the 5 journey stages (we did not recruit from the Not Intending stage). Since no household makes decisions in exactly the same manner as others, we screened for a mix of caregiving styles identified in parenting literature (Table 6) [46, 47].

Table 5. Households sampled for journey mapping.

|  | Actively Aligning | Anticipating | Not Advocating | Advocating | TOTAL |
|---|---|---|---|---|---|
| **Authoritarian** | 3 households (HH) | 3 HH | 3 HH | 3 HH | 12 HH |
| **Authoritative** | 3 HH | 3 HH | 3 HH | 3 HH | 12 HH |
| **Permissive** | 3 HH | 3 HH | 3 HH | 3 HH | 12 HH |
|  |  |  |  |  | **36 HH** |

Table 6. Parenting styles sampled in journey mapping research.

| Parenting Style | Definition |
|---|---|
| **Authoritarian** | Extremely strict: Parents expect their children to follow the rules, with no discussion or compromise. |
| **Authoritative** | Combines warmth, sensitivity, and the setting of limits. Parents use positive reinforcement and reasoning to guide their children. They avoid resorting to threats or punishments. |
| **Permissive** | Parents view their child as their equal. Gift-giving and bribery are their primary parenting tools, rather than setting boundaries and expectations. They place few demands on the children and have a difficult time saying "no," as they avoid asserting authority and confrontation. They also always avoid punishment. |

                                                                      

We hypothesized that household decision-making dynamics would vary based on the dominant caregiving style within the household. As decision-making around adolescent health combines factors like interactions between caregivers and adolescents, family values/rules, and roles, the prior science around caregiving styles was a useful lens to understand the existing context of the household as a unit [47].

Local moderators from each community were trained to conduct journey mapping interviews in local dialect over the course of 2 weeks. Training included the methodology principles, use of discussion guides and the journey mapping tool-kit, respondent interaction, and behavioral probing specific to each journey milestone.

Paired with the discussion guide, the journey mapping tool kit consisted of a paper "journey canvas" (see S2 File) to record the perspective of household participants and capture insights at the transition from one journey stage to the next. Moderators conducted mock journey-mapping interviews with each other and then finally on a small sample of actual households to hone their approach before scaling up to field a full sample.

A screening questionnaire (see S3 File) was used by trained interviewers to identify the caregiving style of each household to ensure that the requisite number of each style was selected for journey-mapping interviews. Questions were derived from "Development and Validation of a Short Form of the Alabama Parenting Questionnaire" [48], contextualized to the Zambian community context. We took three steps in adapting the parenting style survey to the Zambian context: 1. We highlighted phrases we judged as unique in the American (school) context and developed alternative phrasing more suitable for the African/Zambian context. 2. Our local research lead in Zambia re-articulated and translated questions into local context and language. 3. Questions were stress tested in a limited sample pilot study. The questionnaire was translated into local languages, and respondents were recruited door-to-door. The journey mapping took place at participants' homes, but separately from each other to avoid social sensitivity, influence, and bias dynamics.

Each household member was provided with a journey canvas and set of tools to complete their journey with the aid of the moderator. Journey mapping interviews lasted approximately 90 minutes and were audio recorded. Moderators completed a reporting template that captured all key insights gathered in the journey mapping interview. Reports not already in English were translated.

The data was organized into household groups (son, mother figure, father figure) and then divided by stage and analyzed to identify key patterns of behavior, including goals, interactions that seek or avoid journey milestones, perceived implications of interactions, and personal and social influences among all three household members. Analysis data was then codified into normalized behavioral drivers and barriers within each stage of the journey. Tim Sweeney and Jeff Mulhausen (authors) were primary coders for the data. Each has been deploying the underlying methodology for over 10 years. Initially, notes and transcripts were organized and translated by local moderators to interpret and preserve context. The coders then took a thematic analysis approach to analyze the data. Each coder independently coded interviews. After the initial coding round, coders collectively reviewed preliminary codes to refine the framework, address discrepancies and capture key themes. Once the set of codes were finalized, the coders re-coded the interviews to ensure reliability and consistency (For the analysis worksheet, see S4 File.)

Differences in the journeys between authoritarian and permissive caregiving styles were evident, but there were fewer distinctions between either of these and the journey in households with an authoritative caregiving style. We therefore developed two journey pattern indexes describing the key patterns and differences between the authoritarian and permissive households, including the perspectives of all 3 household members (see S5 File). Future interventions could be designed to account for these differences.

### Ethnolab

For the Ethnolab, 150 adolescent males aged 10–19 and 180 caregivers of adolescents were recruited through door-to-door recruitment in Lusaka district (Lusaka province) and Sioma district (Western province), which have high population density and high prevalence of HIV. This sample size was found to consistent with existing literature which have used

mixed-methods [49] and gamification [50] to study HIV prevention, thereby enhancing the credibility of the findings and allowing for a balance between quantitative rigor and qualitative depth. Selection criteria were the age group of the adolescents (10–14 and 15–17, 18–19), and caregivers of adolescents in these age groups, and their alignment to 4 of the 5 journey stages, with the "actively aligning" stage subdivided into a) adolescent not actively aligning but caregiver actively aligning (A- C+), and b) adolescent actively aligning but caregiver not actively aligning (A+ C-) (Table 7). This alignment was determined through a screening tool. An amount of 100 Kwacha (~$5.00 USD) was paid to each participant to compensate them for their time or lost wages, if any.

Ethnolab was used as a research instrument to identify the levers which would enable adolescents and caregivers to overcome barriers faced individually and as a triad. Ethnolab is a proprietary behavioral research methodology grounded in behavioral science that seeks a balance between the purity of ethnographic research and the controlled, experimental nature of a laboratory. It counters the "say-do gap" by simulating the context in which respondents make decisions, so that researchers can then observe them making choices and taking decisions. This helps researchers understand respondents' motivators, barriers, and mental models via observation as well as questioning. To avoid social desirability bias, Ethnolab gamifies the context to help overcome judgments of right and wrong, encouraging respondents to give honest responses. Gamification also makes it easier to build rapport and engagement, while reducing respondent fatigue.

Ethnolab addresses the out-of-context problem by countering the "hot-cold" empathy gap [51]. When we are in the "hot" or emotional state, we do not understand how much it is affecting our behavior, but in the "cold" or rational state, we do not realize how much our decisions would change were we in the hot state. Out-of-context surveys tend to take place in the cold state, but the empathy gap leads respondents to incorrectly predict or account for their hot-state behaviors. The realistic context and game-like context of Ethnolab move respondents into a hot state of decision-making. This makes it possible to observe them taking the decision and immediately thereafter to have a conversation about it, while respondents are still engaged with their emotions and other non-conscious drivers of decision-making. Hot-state discussions thus provide insights which are closer to the respondent's real-world decision-making.

Finally, Ethnolab provides a platform to test interventions or levers to overcome barriers by enabling decision-making in this hot state. Thus, it can be used as a predictive research tool to identify levers for preference reversal.

In the Ethnolab, respondents with similar profiles (age, circumcision status, journey stage) are put in a simulated decision-making context and presented with narrated and illustrated scenarios containing hypothetical situations that

**Table 7. Sample of adolescents and caregivers for Ethnolab.**

**ADOLESCENTS PER PROVINCE**

|  | Not intending | Actively aligning (A- C+) | Actively aligning (A+ C-) | Anticipating | Not advocating | Total |
|---|---|---|---|---|---|---|
| *10-14 years* | 5 | 5 | 5 | 5 | 5 | 25 |
| *15-17 years* | 5 | 5 | 5 | 5 | 5 | 25 |
| *18-19 years* | 5 | 5 | 5 | 5 | 5 | 25 |

**TOTAL ADOLESCENTS in two provinces = 75 x 2 = 150**

**CAREGIVER PER PROVINCE**

|  | Not intending | Actively aligning (A- C+) | Actively aligning (A+ C-) | Anticipating | Not advocating | Total |
|---|---|---|---|---|---|---|
| *10-14 years* | 3 M, 3F | 3 M, 3F | 3 M, 3F | 3 M, 3F | 3 M, 3F | 30 |
| *15-17 years* | 3 M, 3F | 3 M, 3F | 3 M, 3F | 3 M, 3F | 3 M, 3F | 30 |
| *18-19 years* | 3 M, 3F | 3 M, 3F | 3 M, 3F | 3 M, 3F | 3 M, 3F | 30 |

**TOTAL CAREGIVERS in two provinces = 90 x 2 = 180**

mimic the real-life context in which the behavior of interest takes place. The scenarios are aligned with the journey stage of the participants. Each scenario is designed to test an underlying behavioral principle(s) and ends in a decision conundrum with multiple possible outcomes.

For each scenario, participants are asked to guess what *other* participants are likely to do in this scenario, rather than stating what they themselves are likely to do. This helps reduce social-desirability bias. The scenarios are presented in the context of a board game with seven or eight rounds. In each round, if a participant's choice of decision outcomes matches the choice of the majority of the participants, they win the chance to roll a die and move their game piece a corresponding number of places toward the center of the board. After the game, the scenarios are discussed while the participants are in the hot state. The data generated from the responses, along with the ensuing discussion, helps elicit behavioral insights, identifying drivers of decisions and preference reversal.

Each scenario is composed of a narrative and question which describes a specific barrier to be overcome by the adolescent or caregiver or both; and a set of options outlining ways in which the barrier may be overcome, based on behavioral principles. Learnings from the formative research stage, together with prior work in VMMC conducted by the consortium, informed the development of the scenarios. An example of a scenario and corresponding options have been presented in Figs 1 and 2 respectively.

For a table of all the Ethnolab scenarios, options, and responses given, see S6 File.

For the discussion, the adolescents were divided into 2 groups, those in school and those out of school, and caregivers were divided into female and male caregivers. Participants aged 18–19 were divided among those living with their caregivers, and those living independently. The scenario was narrated again, followed by a discussion exploring participants' experiences of facing the barrier and past experiences (their own or others') which prompted them to choose a particular option (lever) during the game. Since circumcision is a one-time decision and the participants had not yet overcome the barriers represented by the scenarios (i.e., they/the adolescents had not yet been circumcised), they were encouraged to share experiences of how others had used the chosen lever to overcome the barrier, or how they themselves had used the lever in a different context.

Moderators for the Ethnolab were local people fluent in local languages (Nyanja and Lozi) who were trained remotely by the consortium research team. Participants gave consent, and the sessions were audio-recorded and transcribed. The transcripts were analyzed using deductive analysis, drawing from Ellsworth & Scherer's emotional appraisal theory [43]

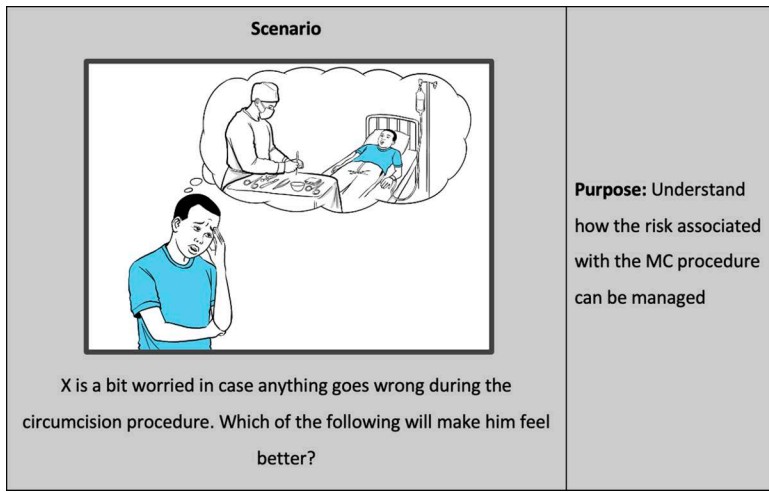

**Fig 1. Sample Ethnolab scenario.**

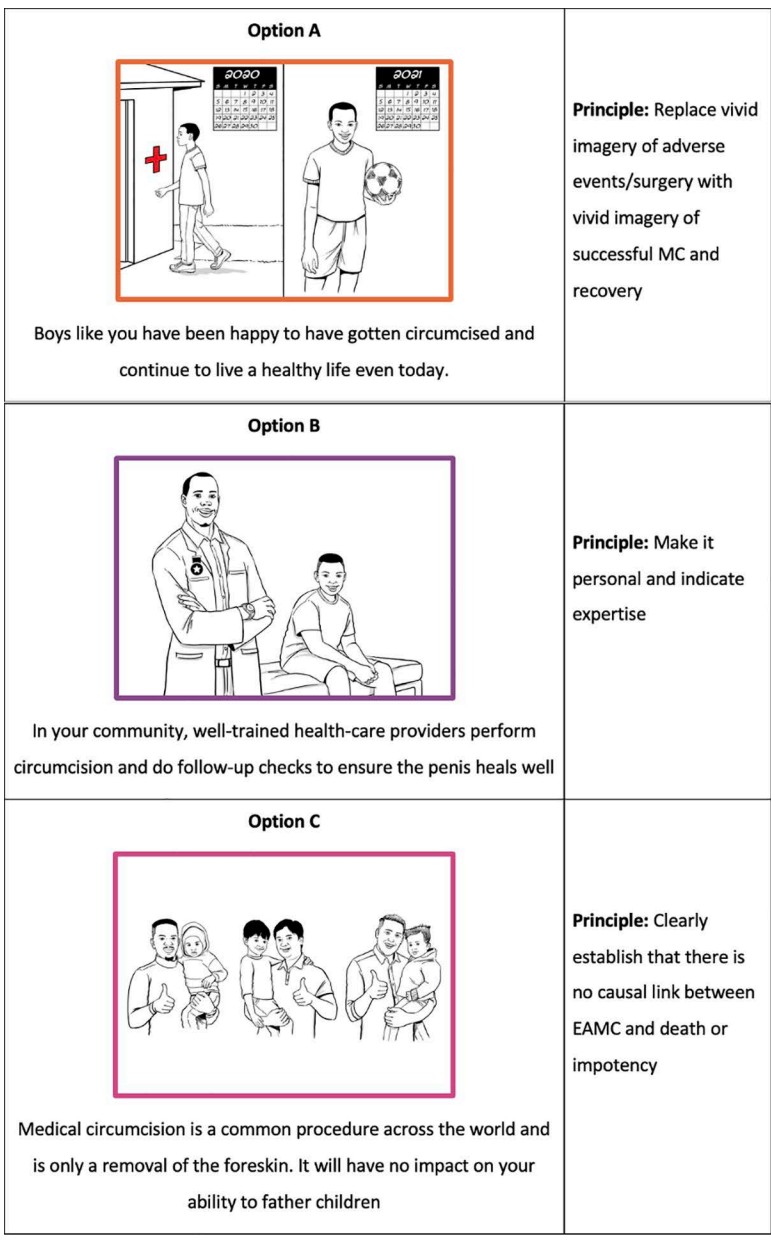

**Fig 2. Sample Ethnolab options.**

and theme matching and pattern identification from our previous research [39]. The authors (Namiya Jain, Rasi Surana and Alok Gangaramany) completed an initial coding of transcripts to systematically identify key themes and dimensions emerging from the data. The dimensions mapped were barrier themes, context, and emotional appraisal elements. The codes from each stage were further analyzed as separate groups.

Game data was analyzed to find similarities and differences between the responses given by caregivers and adolescents, and between different age groups. The game data was further triangulated with data from the discussions. We

focused on how the barriers were experienced, how they were overcome, how households progressed through the journey, and the interactions between the members of the household. The analyzed data was then synthesized into narratives along the following dimensions: a) learnings by journey stage; b) barriers, and c) ways to overcome the barriers.

Finally, the learnings from the journey mapping and the Ethnolab were synthesized through a process of discussion over multiple working sessions, to align on the journey and the barriers faced in each stage and map them onto a solutioning framework (to be published) to allow organizations to easily comprehend household decision dynamics, and optimize existing interventions and/or design new interventions.

The COVID-19 pandemic impacted the initial research plans, and the methodology was adapted to the constraints posed by remote research. Although all travel by the research team was suspended due to travel restrictions and the uncertainty of the pandemic, the team was able to conduct the research, thanks to strong partnerships with in-country teams. As noted, formative research recruitment initially had to be conducted by phone, but reverted to being done face-to-face once lockdown rules were eased. For Ethnolab, the game design and interaction protocols were constructed to ensure the safety of participants and moderators, including social distancing, wearing masks, providing handwashing facilities, frequent sanitizing of the venue and materials, and giving participants sanitizer and masks to take home.

## Results

An overview of the methodological challenges that qualitative research may face and our proposed solutions have been highlighted in Table 8.

Based on the integrated methodology (journey mapping + Ethnolab) that we undertook, we were able to identify nuanced barriers and drivers for VMMC among adolescents, along with actionable levers for change as recognized through an extension of the research process.

The integrated findings of the three phases of research led to the development of a behavioral decision framework. This framework explains the complexity of decisions made at the intersection of caregivers' and adolescents' own goals,

**Table 8. Methodological challenges and proposed solutions.**

| Challenges | How these were addressed |
|---|---|
| Understanding the decision-making dynamic when multiple decision-making agents are involved [9] | **Formative research:** Through this phase, learnings from our previous VMMC research were adapted for EAMC with the aim of understanding the joint decision-making context and socio-cultural influences on the EAMC decisions. Furthermore, data was organized by households to identify hypotheses through triangulation of data points across members of a single household.<br>**Journey mapping:** We organized data based on households so we could jointly understand and analyze the barriers and drivers to decision-making faced by the household unit, i.e., adolescents and caregivers. |
| Say-do gap [52] | **Journey mapping:** Participants were required to fill out their journey canvas highlighting their goals, social influences and journey milestones which moved them forward or held them back in the EAMC journeys.<br>**Ethnolab:** By simulating the decision-making context, the Ethnolab allows researchers to observe the decisions and choices made by respondents. |
| Social desirability bias [29] | **Ethnolab:** In the FGD setting, participants are given a limited time frame to anonymously respond to decision-simulations by guessing what other participants would do, rather than stating what they themselves are likely to do. This takes away the spotlight from the respondent, thereby encouraging more honest responses. |
| Respondent fatigue [32] | **Ethnolab:** The Ethnolab leverages gamified tools to build rapport and engagement with the participants, while reducing respondent fatigue. |
| Hot-cold empathy gap [51] | **Ethnolab:** The decision-simulations are based on relatable, real-life contexts that are constructed based on learnings from prior phases of primary research. These simulations aim to move people closer to a hot state of decision making. |

beliefs, and emotions associated with EAMC. By doing this, we aim to create more actionable and effective strategies for the household as a unit that targets the nuanced, complex decision of EAMC uptake. We present a brief description of the framework, including barriers, drivers, and levers as evidence of the richness of emotionally charged discussions that our methodology enabled. With each phase of our research, we identified key learnings and refined and built on learnings from the previous phase of research. A more detailed account of the study results and the behavioral decision framework will be published separately.

### Stages

Our research and analysis identified 4 distinct stages (Fig 3) anchored to a household's experience of EAMC decision-making, along with certain barrier themes that prevent households from advancing through their decision-making journey.

**Stage 1: Relate.** In this stage, one member of the household unit—either the caregiver or adolescent—has learned of EAMC and has formed an intention to undergo the procedure. We refer to this member as the "initiator," i.e., the household member who initiates the intention for the procedure within the household. To arrive at this intention, the initiator first learns of the procedure (i.e., gains awareness) and then aligns their beliefs with EAMC to avoid any dissonance that might arise from fears, concerns, or mental models that could act as a barrier to intention-building. According to Festinger's cognitive dissonance theory [53], people experience psychological discomfort when holding conflicting thoughts or beliefs and are therefore motivated to modify one of the conflicting beliefs to resolve this [54; see citation for an overview of cognitive dissonance theory]. We found that this alignment process is typically facilitated through information-seeking behaviors and interactions with trusted others, which expand the agent's understanding of EAMC, its benefits, drawbacks, and long-term consequences. Thus, at this stage, the initiator must internalize the value of medical circumcision to develop a strong intention to undergo the procedure.

**Stage 2: Actively align.** Here, the initiator must actively communicate their intention to undergo EAMC with other decision-makers in the household. Conversations about sexual and reproductive health are often uncomfortable, if not taboo, to discuss with family members in African cultures [55], including in Zambia [56]. Given the nature of the topic and the prevalence of patriarchal norms in the culture [57] adolescents and caregivers alike may face challenges when initiating this conversation within the household. Adolescents as initiators must demonstrate agency in introducing and persuading their caregivers, while caregivers as initiators need to assess the adolescent's maturity and contend with potential backlash from close relatives. This is a key stage in joint decision-making, as the journey progresses from a single initiator to a commitment by the household unit to proceed with the adolescent's circumcision.

**Stage 3: Anticipate.** Since the long-term benefits of EAMC have already been internalized by this point, household members no longer question the procedure's benefits. However, the adolescent and caregivers may still grapple with challenges the procedure poses to their near-term goals, such as pain, wound management, and potential loss of wages. Interestingly, while these short-term concerns may have emerged at earlier stages, we found that they often need not

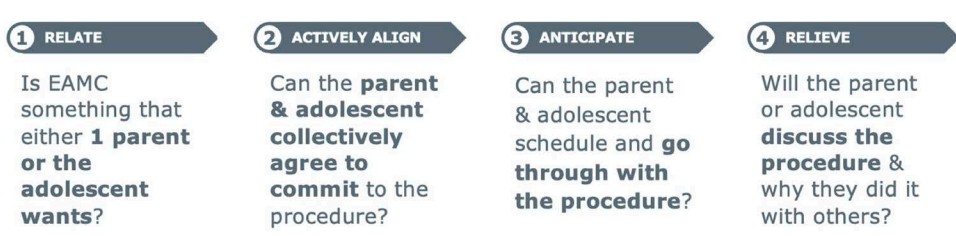

**Fig 3. Four stages through EAMC decision making at the household level.**

be resolved for households to commit to the procedure, as they become more salient closer to actual behavior uptake. Consequently, households may delay the adolescent's circumcision even after committing to it. Intent-to-action gaps are well-studied and documented in public health demand generation literature [see 52 for review]. Addressing these individual, short-term concerns is essential for bridging this intent-action gap and completing the circumcision procedure.

**Stage 4: Relieve.** Descriptive norms—i.e., behaviors that are practiced because they are commonly observed in the community—are important drivers of behavior as they signal what is considered socially acceptable, particularly in contexts of uncertainty [58]. In this stage, while caregivers and adolescents may feel relieved to have completed the procedure, there is often lingering reluctance to share their experience openly with peers. We believe that enabling households to advocate for EAMC can create positive community narratives and perceptions, which is key to generating sustained demand and uptake. Providing adolescents and caregivers with the right communication skills, tools, and opportunities to engage in these conversations will extend the impact of each household's decision to others in the community.

**Barriers**

Several barriers can hinder progress along this journey, while certain drivers can help households move forward. Briefly presented below, these factors represent trade-offs that households encounter at various stages of EAMC uptake. Most barriers and drivers manifest in different ways throughout the EAMC journey, depending on what is relevant at each stage of decision-making.

**1. Anticipation of pain.** Anticipated pain includes the pain of the procedure and pain during the healing process. Adolescents' understanding of VMMC and the associated pain is largely determined by the vivid narratives and lived experiences shared by their peers and elder brothers who have undergone circumcision. Many adolescents have an inherent dislike or fear of injections (which are needed for anesthetic), and they are scared by the idea of having their penis cut with a knife or scissors. A representative quote from a boy was: *"I always imagine how sharp the knife is even when you accidently cut yourself; imagine cutting the foreskin on one's manhood. It would really be painful."*

Upon arriving at the clinic, anticipation of pain may be driven by seeing the surgical equipment laid out, hearing other boys crying during the procedure, and observing how they walk differently immediately afterward; this may cause adolescents to run away from the clinic. *"I saw friends who were crying and I got scared... People were being circumcised and were crying so I thought it was painful and I decided to run away."* The anticipation of pain early in the journey toward circumcision may be overcome by clarifying what the procedure entails and assuring the adolescent that someone close to them will support them as they go through the pain. Caregivers in favor of circumcision can help younger adolescents by asking them what they are afraid of and discussing it to make them feel better. Adolescents require caregivers to communicate confidence that he can go through with the decision, and to provide support immediately after the procedure, such as taking him home in a vehicle. *"I told my son that when you go for circumcision, we will take good care of you. We will be cleaning you and will give you the medication that you will be given at the hospital. At the hospital, they will give you an injection before they start the procedure so that you don't feel any pain. When you come back home, you will not be doing anything until you recover."*

Adolescents can vividly anticipate the pain of healing, based on the experiences of others. These experiences include boys crying in pain, walking with their legs apart, having to stay indoors and wear a *chitenge* (a loose fabric wrap around the waist), and experiencing pain during activities such as bathing, cleaning the wound, urinating, sleeping, or getting erections. They may feel that if their peers and older brothers could not cope with the pain, they will not be able to either. This can cause them to delay going through with circumcision. Caregivers trying to persuade adolescents must overcome these internalized narratives, for example by reassuring them that they will be given painkillers to cope with the pain. As one father said of his adolescent son: *"I had to sit him down and explain all the procedures to him step by step. I first told him that he was going to be given an injection to make the area numb so that he won't be able to feel the pain as they are*

*carrying out the operation. I also told him that after the operation was done, he was going to be given painkillers to carry with him home… I reminded him that he was going to be able to go back to his usual self within a short period of time. That really motivated him a lot."*

A further element is the perception of how well the adolescent can tolerate pain at his specific age. Some female caregivers decide to not circumcise their adolescent because they cannot bear to put them through the pain or see them in pain. They lack clarity themselves on the extent of the pain adolescents experience and would benefit from an effective heuristic to estimate this. For example, a pain-o-meter that could help mothers associate the severity of pain at different stages of the procedure and recovery period with more commonly occurring pains that they can relate with such as headaches, thorn pricks or cuts.

**2. Anticipated loss.** In contexts of uncertainty, people are often willing to undertake risks (such as maintaining the status quo) to avoid perceived losses [59]. This effect is more pronounced among individuals who believe they lack the necessary economic or social resources to manage these losses [59]. The avoidance of anticipated losses is a major barrier holding people back in their EAMC journeys, given the irreversibility of, and uncertainty associated with, the procedure.

Families from traditionally non-circumcising tribes may consider circumcision as a loss of family identity. They may feel they cannot deviate from the path shown to them by their ancestors and must raise their boys in the way they were brought up: *"It's up to us parents to make sure we raise our children according to our origins and tradition."* Adolescents seeking consent from caregivers anticipating this loss may struggle to convince them due to this barrier. Approaches to addressing this concern could include making a clear distinction between medical circumcision and traditional circumcision, emphasizing that making an exception to traditional practice in this one case is not a wholesale rejection of the family's identity.

For younger boys, there may be anticipated loss in the form of the inability to play while they are healing. *"I was scared that maybe when we go to play soccer because we formed a team, I will not play for four weeks."* Older adolescents and caregivers are concerned about the enforced break from school activities or household responsibilities during convalescence. *"One has to have a good choice so that he knows that if I go on this day, I will be able to have enough time to heal, because I will not be the one to herd the cattle."* Caregivers depending on an adolescent for household chores either may not give consent, or may delay circumcision, if they anticipate that their adolescent boy will be unavailable to help while he is healing. This can be overcome if caregivers are helped to plan the procedure for a time when the adolescent can more easily take time off from household chores, or when someone else is available to manage these. *"He chose the date/day because he wanted to herd cattle the first two weeks before going for circumcision so that he could heal before his turn for herding cattle comes."* Caregivers can also be motivated not to delay by highlighting their potential regret if their boy should contract HIV as a result of being uncircumcised.

Older adolescents may also be concerned about lost time at school, or lost wages if they are working. They may also anticipate that sex will be less pleasurable if they are circumcised, while those with a sexual partner may worry about having to abstain from sex during the healing period. Addressing these barriers of anticipated loss would involve reassuring the adolescents that the cessation of sexual activities is for a short time and that the benefits are long-lasting. The barrier of loss of pleasure can be overcome by highlighting women's preference for men who are circumcised, as they perceive them to be desirable, caring, and responsible.

**3. Uncertainty.** Uncertainty refers to a conscious awareness of lacking sufficient information to predict outcomes accurately [60]. Feelings of uncertainty have been shown to result in cognitive distortions [61], negative affect [62], and avoidance behaviors [63]. In the context of EAMC, uncertainty manifests in various ways across the four stages, often preventing adolescents and caregivers from progressing along this journey.

A barrier for some boys is uncertainty about whether their caregivers will give permission for them to be circumcised. Younger boys may be scared of approaching caregivers. *"My mother said, 'I don't know why your father is refusing [your*

*request]; it might be too painful for you' … so I just stopped [asking]."* Older adolescents might address this by reminding their caregivers of other ways in which they have demonstrated maturity and responsibility for significant decisions. Health providers could also help address this barrier by communicating reasons for circumcision that are relevant and reassuring to caregivers. *"My friend just took himself to the hospital, but when he got there he was told to go back home and come back with his parents. But the good part for him was, one of the nurses there knows my friend's mother so they went together and the nurse explained to the mother, and she agreed and took him to get circumcised."*

Conversely, sometimes caregivers who want the adolescent to be circumcised struggle to convince him, because the benefits either seem too distant to him (e.g., future protection from disease) or insufficient to balance the trade-offs. Caregivers often fail to identify and address the adolescent's specific barriers. Presenting circumcision as a topic for discussion and learning, rather than something they insist that the adolescent do, may help caregivers alleviate their adolescent's uncertainty. Involving health providers can help, since older adolescents tend to trust them more than their caregivers for accurate information on health issues. *"The [parents] said if you want to know more let me just take you to the clinic… I am not forcing you to get circumcised but I just want you to hear more about circumcision. He accepted going to the clinic, and they went together, and they took him to the counselors who counseled him very well, and that is how he got circumcised."*

As with distrust, boys and their families are influenced by the experiences of people in their own community. Stories they have heard or experiences they have observed may make them uncertain about whether the circumcision wound will heal properly, or whether it will take longer to heal than it should, and this can deter them from circumcision. *"Those who have been circumcised already, they say, 'Mine delayed to heal,' the other, 'Mine healed fast,' so I would ask myself, what if mine delays to heal?"* For older adolescents, uncertainty about the healing process may become linked to fears of loss of sexual functioning or of death. As with distrust, the experiences of people in their own community influence the concerns of boys and their families. The uncertainty also leads to hesitation in advocating for circumcision. These concerns can be addressed through expert communication about the length of the healing period, how to take care of the wound, and how the adolescent should expect to feel each day during the process.

A final element of uncertainty can be the adolescent's lack of self-efficacy about recommending circumcision to his peers after undergoing the procedure himself: he may feel he lacks the knowledge and authority to talk about it, is uncertain how to raise the topic, or that his peers will not be interested. Approaches to counter this include providing the adolescent and his caregivers on their last visit to the clinic with tools and materials to help them recommend circumcision, along with opportunities to "shadow" others who are already successful advocates. Families that do not wish to advocate directly themselves can be encouraged to direct other individuals to learn more from the health system.

**4. Distrust.** Trust refers to the degree to which one feels they can rely on a person, process, or institution to act in a predictable and dependable manner [64]. Trust plays a key role in bridging information gaps, particularly in contexts of uncertainty and information asymmetry [65]. While trust in the EAMC procedure and the health system is essential for helping household members overcome the uncertainty they face, existing trust barriers currently prevent progress along the EAMC journey.

Some boys and their caregivers are distrustful of the safety of the circumcision procedure and have concerns about the quality of the clinic and the professionals working there. They may worry that a mistake by an inexperienced provider will lead to lasting negative consequences for the boy, such as penile deformity, or an inability to urinate or to father children; the latter is a concern particularly among older adolescents. *"Some boys were saying there are some health workers who do not know how to cut, so if they cut you … your manhood can be destroyed."* These concerns can be allayed by verbal communication from family members, peers, or a doctor that the procedure will be done by experts who have performed many such procedures, that only the foreskin will be removed and that the boy will not be injured or die.

Although most adolescents and their caregivers understand that circumcision helps protect them from HIV, they may be skeptical about this if they know people who were circumcised who went on to contract HIV, or if they are from

non-circumcising traditions but know no one who has contracted the virus. Some adolescents feel that since wearing a condom confers protection from HIV, circumcision is unnecessary. Potential approaches to address this skepticism include acknowledging the importance of wearing a condom during sex even after circumcision, while pointing out that a man may not always remember to do so, and that even condom use does not provide complete protection, which is why the 60% reduction in HIV risk conferred by circumcision is valuable.

**5. Anticipated shame.** Shame is a social emotion that arises from a perceived failure to meet standards set by one's social group [66]. Since shame is associated with negative evaluations of one's social standing, it can lead to negative affect not only toward the situation or action, but also toward oneself as a whole [67]. We found that anticipated shame acts as a deterrent to desired actions across various stages of the EAMC journey.

A significant concern for younger adolescents is the anticipated shame of having to discuss their genitals, or to be naked in front of a female caregiver for wound care; adolescents of all ages feel they are too old to be undressed in front of their caregivers. Caregivers (especially female caregivers) may be similarly embarrassed about tending to their adolescent boys, since it is considered taboo for a female caregiver to see the male child's genitals once he has reached adolescence. Older adolescents find conversations with their female caregivers about sexual matters shameful, though they may be more willing to discuss such topics with the male caregiver. This awkwardness can lead families to delay the decision for circumcision until such time as the adolescent boy can care for the wound himself. To alleviate this concern, caregivers can reassure their male child that they will be taught how to care for the wound themselves, and if necessary, older male siblings or relatives can help wound care, to protect boys from embarrassment with their caregivers. Families can be supported in identifying a helper who the adolescent is comfortable with. *"When I told him about circumcision he asked, how am I going to take care of myself, how will I be able to sit? So I told him that your father will help you, before he goes for work he will clean you and after work. I also told him that if you will allow me, I can also help to clean you."*

In an environment with both circumcised as well as uncircumcised people, circumcised families may anticipate being shamed and mocked for being circumcised. This concern grows as the date for the procedure nears. Boys anticipate being laughed at for having to wear the *chitenge*, which is seen as feminine, or for walking differently, during the healing period. A potential way to address this is to reinforce that most of his peers will support him following the procedure and will say that they are proud to see him wearing the *chitenge* as a sign that he has taken steps to protect himself and the community from HIV. A further source of anticipated shame for older adolescents is the idea that they might be required to undress in front of a female health care provider, or that the surgery would be performed by a woman. Assuring them that they can choose a male health provider can alleviate this concern.

**6. Concerns about lack of ability or responsibility.** The sense of responsibility stems from feeling accountable toward achieving specific intended outcomes through voluntary action [68]. Perceived responsibility serves an important social function. It aids in judging whether a person can be held accountable for an action, with transgressions often leading to punishment or risk of harm [69]. We found that the sense of adolescents' responsibility for undergoing the procedure and ensuring recovery was low among both, caregivers and adolescents themselves. This lack of responsibility hinders households from progressing in their EAMC journeys across the four stages.

Caregivers may feel that their adolescent boys are not mature enough to understand circumcision and its effect on their lives, and as a result they may ignore messaging about it, or avoid or block discussions with the adolescent. Conversely, caregivers who feel their adolescent boy is mature enough to make the decision may avoid discussing it directly because they do not want to impose their views, or because they anticipate feeling guilty if they appear to press him towards a decision that he later regrets. Older adolescents aged 18–19 are legally able to consent to circumcision without their caregiver's agreement but may still look to them for support, especially if they lack the confidence to make a decision that they perceive as carrying risks; however, caregivers may also regard their adolescent as too immature to make the decision, especially if he is still living with them. Having community mobilizers remind caregivers of the adolescent's ability

 

to take responsibility in age-appropriate ways but also of the importance of continuing to support his health decisions, may address these concerns.

Caregivers of younger adolescent boys may be concerned that their boy is not old enough to care for the wound properly during the healing period. The boys are also concerned about their self-efficacy for wound care, especially if they have heard stories of other boys who forgot to wash the wound or did so incorrectly. This leads them to believe that there is a high chance they will do the same. "*I saw my friends were failing to clean themselves, so their fathers and uncle were the ones cleaning them, so I would ask myself, can I manage to clean my wound if my friends are failing to do it on their own? So I developed that fear not to do it.*" Caregivers may distrust their adolescent's ability to refrain from playing or other activities that are not allowed during the healing period. In the case of older adolescents, female caregivers are unwilling to tend the wound because of taboos about seeing their adolescent boy naked, yet they worry that he will be unable to manage the wound care adequately himself. Caregivers and their adolescent boys need reassurance that they will be able to manage, and to be given instructions on wound care. Younger adolescents need reassurance that they will not have to cope on their own, and older adolescents that they will be able to manage it themselves. "*When [my son] was refusing to go for circumcision, he was telling me that he will not be able to do wound care. So as a way of convincing him, I told him that he won't have to worry because I will help him with everything.*" "*My son refused [to be circumcised], saying I cannot manage to handle this wound on my own… I told him that I will find a friend who has been circumcised so that he can also tell you how they managed.*"

**7. Anticipated conflict.** Studies have found that conflict avoidance or minimization is an important goal driving (or inhibiting) action in collectivist cultures – such as those in Africa – compared to individualistic and honor-driven cultures [68]. This becomes particularly relevant in the EAMC context in Zambia where there are power and status differentials exist within households and communities [70].

Adolescent boys may avoid discussing EAMC with their caregivers if they anticipate that either or both will be upset. Reassurance from peers and help finding the best opportunity to initiate a discussion can alleviate this barrier. Some older adolescents anticipate conflict if they get circumcised without first consulting their caregivers, especially if there were to be medical complications. A potential approach to addressing this would include building their capacity to address their caregiver's fears (e.g., anticipated loss of family identity, uncertainty around wound care responsibility, child's maturity in handling the procedure) to help them discuss the issue with parents beforehand. It is also important to build older adolescents' confidence that their caregivers will trust their decision if he shows that he understands what he is consenting to.

Once an adolescent is circumcised, there may also be an expectation of conflict if they recommend circumcision to another person who goes on to have an adverse experience. This can deter a circumcised adolescent – or his family – from advocating for circumcision among his peers or in the general community. As a parent described it, "*The problem with persuading other people to get circumcised [is that] if anything goes wrong, one can be blamed for having been the one to have persuaded them against their wishes. That's why I just kept my secret with my son after getting him circumcised.*" Ensuring that caregivers and the adolescent do not bear all the responsibility of advocating for circumcision, but that it is shared with trained mobilizers and healthcare professionals, could help ensure that blame for any adverse experience is not directed at the families.

### Enablers for circumcision

We noted several enabling factors for circumcision. In-group seeking, i.e., the desire to be similar and part of a group of friends, motivates the adolescent to seek circumcision. "*I am no longer afraid of getting circumcised … It is because all my friends in our group are circumcised except for me.*" Caregivers are also motivated towards circumcision when there are many circumcising families in their community. Some older adolescents are motivated to choose circumcision precisely because they are teased by their age peers for not being circumcised. For older adolescents, key motivators include being appreciated by women for having taken responsibility for one's sexual health, as well as the

perception of improved sexual attractiveness and better sexual performance after circumcision. For some caregivers, better hygiene is a reason for having their adolescent boys circumcised. Caregivers of younger adolescents tend to believe that the healing process is faster at a younger age, since younger adolescents have fewer responsibilities requiring physical activity, and it is easier to control the adolescent boy during the healing period. Related to this is a concern about the cost of inaction if the adolescent were to contract HIV or another sexually transmitted infection as a result of not being circumcised. This is particularly the case for caregivers of older adolescents who are (or may shortly become) sexually active. This fear can create a sense of urgency that overcomes any tendency to procrastinate on the decision. Finally, the desire of families to be liked and appreciated by their community can motivate them to seek circumcision for their adolescent, if this is something that is valued and supported by the community. Institutions such as the church – an arbiter of respect and esteem in the community – can reinforce this motivation by explicitly supporting circumcision.

## Discussion

While EAMC is not a new strategy, programs have historically focused on persuading the individual to undergo circumcision. For boys who have not reached legal adulthood (18 years in most countries), given the need for caregiver consent as well as the adolescent's assent, this simplistic approach does not take full account of the boy's context and relationships. It is reaching the limits of its effectiveness, and new strategies are required to reach targets for EAMC.

Our work builds an understanding of EAMC as a group decision dynamic made among the triad of adolescent, mother figure and father figure. It takes account of the differing barriers faced by the adolescent boy and by his caregiver, and how the concerns of each can be effectively addressed. In families of younger adolescents, EAMC is not treated like other decisions, and caregivers seek the agreement of the adolescent, since his cooperation is necessary for a successful outcome. It is an active negotiation between the younger adolescent and his caregiver. Among 18–19-year-olds, a range of caregiver involvement is seen: some adolescents feel they cannot make the decision on their own, some feel they must involve their caregiver even if they make the decision for themselves, while others make the decision independently. Often, their living situation influences their VMMC decision-making. In families where older adolescents continue to live with their caregivers, they tend to have lower decision-making power, find it challenging to seek alignment with their caregiver, and the caregivers have better opportunities to influence the adolescent boy's decisions. By contrast, caregivers of older adolescents who are living independently feel a reduced sense of control, as they cannot easily know or predict their adolescent's thoughts and actions.

From a programmatic standpoint, there are three main conclusions. First, the demand generation strategy for EAMC needs to be rethought at the level of a family unit by bringing caregivers into the conversation, in addition to engaging with adolescents. Programs need to progress from targeting adolescents – for example through school-based initiatives where caregivers are excluded from the intention-building and decision-making processes – to more community-driven programs that encourage conversation within (and among) households. Second, power structures, decision-making dynamics and barriers faced by the household continue to evolve as adolescents move across the 10–19 age band. It becomes imperative to approach EAMC demand generation based on nuanced barriers and needs of the household at different points in the adolescent's and caregivers' journey. This could entail differentiated outreach to households depending on whether younger or older adolescents are being targeted, and aiding community outreach workers in identifying the relevant barriers to be addressed. Third, we find that the initiator of positive EAMC attitudes into households differs from household to household – in some cases it may be the caregivers who build intention around EAMC first, whereas in other cases it could be the adolescent himself. Strategies need to be devised to build trust and engagement between members of the household to enable conversation for more holistically informed judgements on the risks and benefits of undergoing EAMC. This is especially important in instances where adolescents or caregivers who feel like they are not in control of

the EAMC decision must convince other household members to overcome their fears associated with the procedure and post-procedure care.

## Conclusion

The family as a whole perceives EAMC to be a risky decision with uncertain consequences. Our study was able to draw out nuanced barriers and levers for different actors involved in this decision at different stages of the proposed EAMC journey. An integrated approach to data collection, analysis and synthesis was important. The methodologies leveraged played dual roles in contributing to understanding adolescents' EAMC journey: 1) Journey mapping was integral to the development of the proposed journey and associated barrier identification, and 2) Ethnolab further shaped the barriers and identified strong levers to help guide future attempts to change household behaviors. The integration of these approaches allowed a nuanced understanding of the various actors and factors that come into play during the EAMC decision-making process and enabled a credible perspective on how these could be tackled to guide adolescents and households forward in their EAMC journey.

## Supporting information

**S1 File. Zambia adolescent stats.**
(PPTX)

**S2 File. Journey canvas.**
(PDF)

**S3 File. Journey screener.**
(DOCX)

**S4 File. Journey analysis worksheet.**
(XLSX)

**S5 File. Journey pattern indexes.**
(XLSX)

**S6 File. Ethnolab scenarios and responses.**
(XLSX)

## Acknowledgments

James Baer assisted with the editing and proofreading of the manuscript.

## Author contributions

**Conceptualization:** Rasi Surana, Ram Prasad, Namiya Jain, Alok Gangaramany, Tim Sweeney, Jeff Mulhausen, Steve Kretschmer, Tina Chisenga.

**Formal analysis:** Rasi Surana, Alok Gangaramany, Tim Sweeney, Jeff Mulhausen.

**Funding acquisition:** Alok Gangaramany.

**Investigation:** Rasi Surana, Namiya Jain, Jeff Mulhausen, Alice Nanga.

**Methodology:** Rasi Surana, Namiya Jain, Alok Gangaramany, Tim Sweeney, Jeff Mulhausen.

**Project administration:** Rasi Surana, Mothi Prasad, Tim Sweeney, Jeff Mulhausen, Alice Nanga.

**Resources:** Jeff Mulhausen.

**Supervision:** Rasi Surana, Ram Prasad, Namiya Jain, Mothi Prasad, Alok Gangaramany, Tim Sweeney, Jeff Mulhausen, Tina Chisenga.

**Visualization:** Tim Sweeney, Jeff Mulhausen.

**Writing – original draft:** Rasi Surana, Namiya Jain, Jeff Mulhausen.

**Writing – review & editing:** Rasi Surana, Namiya Jain, Alok Gangaramany, Aishwarya Shashi Kumar, Tim Sweeney, Jeff Mulhausen, Steve Kretschmer, Alick Samona, Tina Chisenga.

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
