## [Decision Letter · Decision Letter 0]

1 Aug 2024

PONE-D-23-41791A qualitative research methodology to understand the drivers of the decision dynamic among adolescent boys and their parents/guardians affecting uptake of voluntary medical male circumcision for HIV prevention in ZambiaPLOS ONE

Dear Dr. Gangaramany,

Thank you for submitting your manuscript to PLOS ONE. After careful consideration, we feel that it has merit but does not fully meet PLOS ONE’s publication criteria as it currently stands. Therefore, we invite you to submit a revised version of the manuscript that addresses the points raised during the review process.

We look forward to receiving your revised manuscript.

Kind regards,

Hamufare Dumisani Dumisani Mugauri, Ph.D. Public Health

Academic Editor

PLOS ONE

2. In the ethics statement in the Methods, you have specified that verbal consent was obtained. Please provide additional details regarding how this consent was documented and witnessed, and state whether this was approved by the IRB

Reviewers' comments:

Reviewer's Responses to Questions

**Comments to the Author**

1. Is the manuscript technically sound, and do the data support the conclusions?

Reviewer #1: Partly

Reviewer #2: Partly

Reviewer #3: Partly

2. Has the statistical analysis been performed appropriately and rigorously? 

Reviewer #1: N/A

Reviewer #2: N/A

Reviewer #3: N/A

3. Have the authors made all data underlying the findings in their manuscript fully available?

Reviewer #1: No

Reviewer #2: Yes

Reviewer #3: No

4. Is the manuscript presented in an intelligible fashion and written in standard English?

Reviewer #1: Yes

Reviewer #2: No

Reviewer #3: No

5. Review Comments to the Author

Reviewer #1: Lessons from the investigations presented in the paper are helpful and this deep dive into the drivers and barriers to circumcision is informative. Insightful lessons are presented in the results section but there is lack of linkage with the methods and the data; that is, it was not clear how or from where the categories were derived (interviews, journey mapping, and/or ethnolab?). Nonetheless, with some revision, this research should be published somewhere.

Here are comments about specific sections/content:

The authors claimed that VMMC presents the greatest possible protection but does not specify against what. It is safe to assume against HIV. What about PrEP and condoms?

Specify for whom the 15 VMMC countries are priority (WHO, UNAIDS, PEPFAR?)

The statement, “VMMC is relevant for adolescents because it is a one-time intervention that confers the greatest possible protection if it takes place before their sexual debut” is misleading. VMMC does not provide the “greatest possible protection” but it is the only one-time intervention with partial protection since other prevention methods like condoms and PrEP require consistent use to be effective.

A few misplaced commas for lines 50-54 makes the sentence difficult to follow. Additionally, the statement that, “Since 2020 the World Health Organization (WHO) has recommended that VMMC programs for

adolescents focus on those aged 15-19, because of concerns about a higher rate of some serious

adverse events among adolescents ages 10-14 years,” needs citation.

72 The authors use a “behavioral decision making framework” but a brief description is needed about this framework because it seems to be the foundation of this paper.

132 Add citation to, “Decisions are driven by emotions, by mental shortcuts (heuristics) and other non-conscious drivers.”

A better description of Immersion method is needed. It sounds like interviews but this isn’t directly stated. If this is the case, why not just label it interviews? The analytical method for interview data was unclear. It sounds like the stages derived were based on previous experience and from findings from the current interviews but how the Immersion findings supported stages derived from “previous experience” was unclear.

At the beginning of the Methods section, brief explanation of Immersion, Journey Mapping, Ethnolab is needed. It was difficult to follow the authors’ point about each of these because the description did not come until later in the paper.

Time orientation for each of the data collection methods is needed. When were data for Immersion, Journey Mapping, Ethnolab collected and how long did it take ?

The authors “hypothesized that household decision-making dynamics would vary based on the dominant parenting style within the household” but not much background justification was presented on this “hypothesis.” Additionally, the authors used the Alabama Parenting Questionnaire to assess parenting style and stated to have “contextualized to the Zambian community context.” A description of how they “contextualized” this measure is needed (e.g., back translated, revised the items?).

The analytical plan needed more details (e.g., number of coders and their background, software used for analysis, the role of the authors). It would be helpful for authors to go through the Consolidated criteria for reporting qualitative studies (COREQ) to ensure that critical information have been presented.

This paper mentioned multiple times the triad (i.e., adolescent, mother, father). Was there ever a case in which the unit was not a triad or had different members (single parent households or non-parent caregiver). If not, how might this lack of representation limit the generalizability of the findings to certain types of families?

Reviewer #2: This paper requires some significant work to pass for publication. Here are my key comments:

1. Methods: The methods mention immersions as a method and describe this as in-depth interviews. This is incorrect. Immersions are a lot of in-depth and include spending time in participant lives. This should be corrected. Further the recruitment criteria should separate adolescent boys from parents since parents do not really fit into circumcision status. Also, you further segment parents in parenting style but you donot mention this as a criteria fror recruitment of parents. Lastly, given that decision making reduces with age, should the study not have interviewed more of the younger ages i.e. below 15 years?

For ethics approval it standard to mention approval number if this is provided by the board used.

2. The work proposes to proffer a behavioral decision making framework. This is not given in the results. This is a key outcome of this work and due to this weakness this work is falling short or may need to be recasted.

3. Related to above, the results section and the title do not fully align. Results focus on barriers and less on decision making. While barriers a underlying to decisions this linkage is not fully examined. The study then becomes more of a traditional barriers and motivators to circumcisions. These are well documented in literature and thus the study will not be adding new knowledge. This linkage needs to be established for the work to be publishable. I find this to be a key comment of my review.

Reviewer #3: • There seems to be too much content compressed into this manuscript, making it difficult to exhaust refined descriptions of each methodological aspect. Thus, there are some gaps in the methodological descriptions.

• I would recommend that authors consider splitting the manuscript into 2 or 3 sizeable and refined manuscripts. For example, each of the 3 elements: the interviews, journey mapping and ethnolab can constitute a distinct manuscript, where all the refined details are included, along with rationales

See attachment for details

6. PLOS authors have the option to publish the peer review history of their article (what does this mean? ). If published, this will include your full peer review and any attached files.

**Do you want your identity to be public for this peer review?** For information about this choice, including consent withdrawal, please see our Privacy Policy .

Reviewer #1: No

Reviewer #2: No

Reviewer #3: **Yes: ** Charles Maibvise

---

## [Author Response · Author response to Decision Letter 1]

24 Jan 2025

The detailed response is in rebuttal letter. Also, included in the content below

1. Is the manuscript technically sound, and do the data support the conclusions?

Reviewer #1: Partly

Reviewer #2: Partly

Reviewer #3: Partly

2. Has the statistical analysis been performed appropriately and rigorously?

Reviewer #1: N/A

Reviewer #2: N/A

Reviewer #3: N/A

3. Have the authors made all data underlying the findings in their manuscript fully available?

Reviewer #1: No

Reviewer #2: Yes

Reviewer #3: No

4. Is the manuscript presented in an intelligible fashion and written in standard English?

Reviewer #1: Yes

Reviewer #2: No

Reviewer #3: No

5. Review Comments to the Author

Reviewer #1:

Lessons from the investigations presented in the paper are helpful and this deep dive into the drivers and barriers to circumcision is informative. Insightful lessons are presented in the results section but there is lack of linkage with the methods and the data; that is, it was not clear how or from where the categories were derived (interviews, journey mapping, and/or ethnolab?). Nonetheless, with some revision, this research should be published somewhere.

Here are comments about specific sections/content:

• The authors claimed that VMMC presents the greatest possible protection but does not specify against what. It is safe to assume against HIV. What about PrEP and condoms?

o Rebuttal: Edited to address feedback and introduce clarity in communication

o Lines: 47 - 54

o Edits: VMMC is a one-time intervention, unlike Pre-exposure prophylaxis (PrEP) or condoms, that significantly reduces the risk of HIV transmission and other sexually transmitted infections (STIs ) [1,2,3]. It is particularly relevant for adolescents as the procedure offers the greatest protection if it is done before sexual debut, as it reduces the lifetime risk of acquiring HIV. Therefore, targeting adolescent boys who have not yet become sexually active maximizes the health benefits of VMMC. VMMC is relevant for adolescents because it is a one-time intervention that confers the greatest possible protection if it takes place before their sexual debut.

• Specify for whom the 15 VMMC countries are priority (WHO, UNAIDS, PEPFAR?)

o Rebuttal: Edited to include requested information

o Lines: 44

o Edits: VMMC has been a key part of HIV prevention interventions in 15 priority countries in Southern and Eastern Africa as identified by UNAIDS & WHO [4].

• The statement, “VMMC is relevant for adolescents because it is a one-time intervention that confers the greatest possible protection if it takes place before their sexual debut” is misleading. VMMC does not provide the “greatest possible protection” but it is the only one-time intervention with partial protection since other prevention methods like condoms and PrEP require consistent use to be effective.

o Rebuttal: Edited to address feedback and introduce clarity in communication

o Lines: 47 - 54

o Edits: VMMC is a one-time intervention, unlike Pre-exposure prophylaxis (PrEP) or condoms, that significantly reduces the risk of HIV transmission and other sexually transmitted infections (STIs ) [1,2,3]. It is particularly relevant for adolescents as the procedure offers the greatest protection if it is done before sexual debut, as it reduces the lifetime risk of acquiring HIV. Therefore, targeting adolescent boys who have not yet become sexually active maximizes the health benefits of VMMC. VMMC is relevant for adolescents because it is a one-time intervention that confers the greatest possible protection if it takes place before their sexual debut.

• A few misplaced commas for lines 50-54 make the sentence difficult to follow.

o Rebuttal: Edited to break up sentence into shorter, more concise sentences to improve clarity in communication

o Lines: 58 - 63

o Edits: In 2020 the World Health Organization (WHO) revised the age criteria for VMMC programs for adolescents (also referred to as Early Adolescent Male Circumcision) to 15-19 years [6] because of concerns about 10-14 year old adolescents’ ability to give informed assent to a medical procedure that is non-urgent and irreversible and due to a higher rate of serious adverse events following Male Circumcision (MC) among this age group. In line with this updated guideline, Zambia’s VMMC program now focuses on reaching uncircumcised males aged 15 and above [7].

• Additionally, the statement that, “Since 2020 the World Health Organization (WHO) has recommended that VMMC programs for adolescents focus on those aged 15-19, because of concerns about a higher rate of some serious adverse events among adolescents ages 10-14 years,” needs citation.

o Rebuttal: Citations added for edited text

o Lines: 59 and 63

o Edits: Same citation for whole paragraph - [Reference #6 in paper - World Health Organization. Policy brief: preventing HIV through safe voluntary medical male circumcision for adolescent boys and men in generalized epidemics: recommendations and key considerations. Geneva: The Organization; 2020.]

• 72 The authors use a “behavioral decision making framework” but a brief description is needed about this framework because it seems to be the foundation of this paper.

o Rebuttal: We have edited the manuscript to provide clarity regarding the objectives of this paper, which is to describe the qualitative methodology undertaken by the authors that aimed to address challenges that traditional qualitative research faces, especially in the context of EAMC decision making. While the behavioural decision making framework was the output of our research, detailing it out falls out of the scope of this paper (as outlined in Line 503).

o Lines: 73 - 84

o Edits: The present paper describes a methodology specially designed to address the complexity underlying decision-making related to voluntary medical male circumcision (VMMC) for adolescents with their parents/guardians in Zambia. By focusing on the joint decision-making dynamics within families and identifying the drivers and barriers to circumcision for adolescents aged 10-19, our methodology provided unique insights into how interventions could be more effectively tailored to increase VMMC uptake. Our study included younger adolescents (10-14 years in addition to 15-19 years) as the study design and initial fieldwork was completed prior to Zambia’s alignment with the revised WHO guidelines promoting VMMC for those aged 15 and older [6]. This paper details the theoretical foundations and qualitative research methods we employed, highlighting how these approaches successfully addressed limitations in traditional research methodologies, ultimately providing actionable insights and a behavioral decision-making framework for improving VMMC uptake (a more detailed account of which will be published separately).

• 132 Add citation to, “Decisions are driven by emotions, by mental shortcuts (heuristics) and other non-conscious drivers.”

o Rebuttal: Citation added

o Lines: 150

o Edits: [27] - Lerner JS, Li Y, Valdesolo P, Kassam KS. Emotion and Decision Making. Annual Review of Psychology [Internet]. 2015;66(1):33.1–25. Available from: https://www.annualreviews.org/content/journals/10.1146/annurev-psych-010213-115043]

• A better description of Immersion method is needed. It sounds like interviews but this isn’t directly stated. If this is the case, why not just label it interviews? The analytical method for interview data was unclear. It sounds like the stages derived were based on previous experience and from findings from the current interviews but how the Immersion findings supported stages derived from “previous experience” was unclear.

o Rebuttal:

1. We have re-named "immersion" as "formative research" across the document, and have called out that it consisted of semi-structured in-depth interviews

2. We have included additional information on our analysis method for the formative research

3. We have included text to inform the linkage between findings from previous research and the formative research

o Edits:

1. Lines 194 - 197 – 1. The formative research consisted of semi-structure in-depth interviews with adolescent boys aged 10-17 (hereafter referred to as younger adolescents) that, along with previous research undertaken by the consortium [Vaish et al., 2015], informed the design of the journey mapping and Ethnolab.

2. Lines 269 – 277 - 2. Transcripts of the interviews were analyzed using deductive analysis, drawing from Ellsworth & Scherer's emotional appraisal theory [43] and thematic learnings from our previous research [39]. The authors (Namiya Jain, Rasi Surana and Alok Gangaramany) coded transcripts to systematically identify key themes and appraisal dimensions emerging from the data. Data from each household were organized and analyzed together, allowing us to examine interviews across all members of a single household in context of one another. This approach enabled more effective triangulation of insights. After coding, the data were synthesized by respondent category and further grouped by age band to capture nuanced patterns across different demographic groups.

3. Lines 278 - 281 - 3. The formative research adapted hypotheses from the previous VMMC research for the joint decision-making context of EAMC to arrive at the stages of a households journey to EAMC (Table 4). This was used as a basis for further stages of research in the study.

• At the beginning of the Methods section, brief explanation of Immersion, Journey Mapping, Ethnolab is needed. It was difficult to follow the authors’ point about each of these because the description did not come until later in the paper.

o Rebuttal: We have included a table detailing out the time frame, sample and objectives of each phase of research to help improve understanding and readability of this section

o Lines: 224

o Edits: Table 2

• Time orientation for each of the data collection methods is needed. When were data for Immersion, Journey Mapping, Ethnolab collected and how long did it take ?

o Rebuttal: The time orientation for each phase of research has been introduced in the process overview table

o Lines: 224

o Edits: Table 2

• The authors “hypothesized that household decision-making dyna mics would vary based on the dominant parenting style within the household” but not much background justification was presented on this “hypothesis.” Additionally, the authors used the Alabama Parenting Questionnaire to assess parenting style and stated to have “contextualized to the Zambian community context.” A description of how they “contextualized” this measure is needed (e.g., back translated, revised the items?).

o Rebuttal: Details added to address feedback

o Edits:

1. Lines: 323 – 326 - As decision-making around adolescent health combines factors like interactions between parents and adolescents, family values/rules, and roles, the prior science around parenting styles was a useful lens to understand the existing context of the household as a unit [47].

2. Lines: 345 - 349 - We took three steps in adapting the parenting style survey to the Zambian context: 1. We highlighted phrases we judged as unique in the American (school) context and developed alternative phrasing more suitable for the African/Zambian context. 2. Our local research lead in Zambia re-articulated and translated questions into local context and language. 3. Questions were stress tested in a limited sample pilot study.

• The analytical plan needed more details (e.g., number of coder s and their background, software used for analysis, the role of the authors). It would be helpful for authors to go through the Consolidated criteria for reporting qualitative studies (COREQ) to ensure that critical information have been presented.

o Rebuttal: Details added across all research stages to address feedback

o Edits:

1. Lines 269 – 277 - Transcripts of the interviews were analyzed using deductive analysis, drawing from Ellsworth & Scherer's emotional appraisal theory [add citation] and thematic learnings from our previous research [add citation]. The authors (Namiya Jain, Rasi Surana and Alok Gangaramany) coded transcripts to systematically identify key themes and appraisal dimensions emerging from the data. Data from each household were organized and analyzed together, allowing us to examine interviews across all members of a single household in context of one another. This approach enabled more effective triangulation of insights. After coding, the data were synthesized by respondent category and further grouped by age band to capture nuanced patterns across different demographic groups.

2. Lines 362 – 369 - Tim Sweeney and Jeff Mulhausen (authors) were primary coders for the data. Each has been deploying the underlying methodology for over 10 years. Initially, notes and transcripts were organized and translated by local moderators to interpret and preserve context. The coders then took a thematic analysis approach to analyze the data. Each coder independently coded interviews. After the initial coding round, coders collectively reviewed preliminary codes to refine the framework, address discrepancies and capture key themes. Once the set of codes were finalized, the coders re-coded the interviews to ensure reliability and consistency.

3. Lines 457 – 463 - The transcripts were analyzed using deductive analysis, drawing from Ellsworth & Scherer's emotional appraisal theory [43] and theme matching and pattern identification from our previous research [Vaish et al., 2015]. The authors (Namiya Jain, Rasi Surana and Alok Gangaramany) completed an initial coding of transcripts to systematically identify key themes and dimensions emerging from the data. The dimensions mapped were barrier themes, context, and emotional appraisal elements. The codes from each stage were further analyzed as separate groups.

• This paper mentioned multiple times the triad (i.e., adolescent, mother, father). Was there ever a case in which the unit was not a triad or had different members (single parent households or non-parent caregiver). If not, how might this lack of representation limit the generalizability of the findings to certain types of families?

o Rebuttal: Clarification about the inclusion of non-parent caregivers and single parent households included. Furthermore, to streamline language across the document, references of son and parents/guardians have been changed to adolescent and caregivers.

o Lines: 211 - 213

o Edits: Across the 3 phases of research, data was collected from adolescents and caregivers (including parent and non-parent caregivers and single-parent households) to understand the joint-decision making dynamics at play for EAMC decisions.

Reviewer #2:

This paper requires some

---

## [Editor Report · Decision Letter 1]

4 Feb 2025

A qualitative research methodology to understand the drivers of the decision dynamic among adolescent boys and their parents/guardians affecting uptake of voluntary medical male circumcision for HIV prevention in Zambia

PONE-D-23-41791R1

Dear Dr. Alok,

We’re pleased to inform you that your manuscript has been judged scientifically suitable for publication and will be formally accepted for publication once it meets all outstanding technical requirements.

Kind regards,

Hamufare Dumisani Mugauri, Ph.D. Medicine and Health Sciences

Academic Editor

PLOS ONE
---

## [Editor Report · Acceptance letter]

PONE-D-23-41791R1

PLOS ONE

Dear Dr. Gangaramany,

I'm pleased to inform you that your manuscript has been deemed suitable for publication in PLOS ONE. Congratulations! Your manuscript is now being handed over to our production team.

Kind regards,

on behalf of

Mr Hamufare Mugauri

Academic Editor

PLOS ONE